# Effect of 8-Week β-Alanine Supplementation on CRP, IL-6, Body Composition, and Bio-Motor Abilities in Elite Male Basketball Players

**DOI:** 10.3390/ijerph192013700

**Published:** 2022-10-21

**Authors:** Ioan Turcu, Bogdan Oancea, Mihaela Chicomban, Gabriel Simion, Sorin Simon, Codruța Ioana Negriu Tiuca, Mircea Nicolae Ordean, Alexandru Gabriel Petrovici, Nicoleta Adina Nicolescu Șeușan, Petronela Lăcrămioara Hăisan, Ioan Teodor Hășmășan, Alexandru Ioan Hulpuș, Iulian Stoian, Cătălin Vasile Ciocan, Ioana Maria Curițianu

**Affiliations:** 1Department of Motor Performance, Faculty of Physical Education and Mountain Sports, Transilvania University of Brasov, 500036 Brasov, Romania; 2Department of Physical Education and Sport, 1 Decembrie 1918 University of Alba Iulia, 510009 Alba Iulia, Romania; 3Department of Environmental Sciences Physics Physical Education and Sport, Faculty of Science, Lucian Blaga University of Sibiu, 550024 Sibiu, Romania; 4Department of Physical Education and Sports Performance, Faculty of Physical Education, Sport and Health Sciences, Vasile Alecsandri of Bacau, 600115 Bacau, Romania; 5Department of Physical Education and Special Motricity, Faculty of Physical Education and Mountain Sports, Transilvania University of Brasov, 500036 Brasov, Romania

**Keywords:** beta-alanine, pro-inflammatory, youth sports, performance, basketball

## Abstract

The purpose of this study was to evaluate the effect of 8-week β-alanine supplementation on C-Reactive Protein (CRP), interleukin-6 (IL-6), body composition, and bio-motor abilities in elite male basketball players. Twenty male basketball players (age: 23 + 0.6 years; body mass: 78.3 + 4.8 kg; height:185.3 + 5.4 cm, %BF, 15.2 ± 4.8) volunteered to participate in this study. They were divided into a β-alanine group (BG, N = 10) and a placebo group (PG, N = 10). All players were preparing for university competitions and had played for over five years. Players used 6.4 g/d of β-alanine in BG and maltodextrin in PG. The participants were involved in regular basketball training three months before the study. CRP, IL-6, body composition parameters, and bio-motor abilities were measured before starting the exercises and after completing the eight-week training period. The research findings showed a significant decrease in CRP and IL-6 and an increase in anaerobic peak power between the pre-test and post-test, as well as between BG and PG groups (*p* < 0.05). Although the other measured factors were a relative improvement compared to the pre-test and also compared to PG, these changes were not statistically significant (*p* < 0.05). Eight weeks of β-alanine supplementation ameliorated increases in IL-6 and CRP associated with in-season physical stressors in collegiate basketball players. These changes in pro-inflammatory cytokines suggest that β-alanine supplementation may be a useful nutritional strategy for immune regulation and can also improve anaerobic performance compared to PG.

## 1. Introduction

Basketball is a multi-sprint sport that includes phases of a high-intensity activity such as sprinting, running, and jumping, less intense movements such as jogging and walking, and active or passive recovery [1]. According to studies, basketball players can run several kilometers throughout a game, including several high-speed forward and lateral movements and decelerations from frequent sprint efforts [2,3]. Each basketball player can make more than 50 vertical jumps per game [3]. In the men’s elite competition, the maximum heart rate is up to 191 beats per minute, and the average heart rate in these players is 171 beats per minute, or 91% of the maximum heart rate [2]. Data from the blood lactate levels of elite male and female basketball players show that anaerobic metabolism plays a significant role in providing energy in a basketball game. The mean blood lactate levels for male basketball players were 8.5 ± 3.1 mmol/L, and the mean for international women’s games was 5.7 ± 2.1 mmol/L. Many activities in basketball are conducted at near-maximum intensities and with anaerobic capability [1,2]. Most play-related activities, on the other hand, are carried out at low to medium intensity (i.e., 40 to 50%) and are followed by recovery through aerobic energy pathways. Aerobic fitness may be more significant in recovery during frequent high-intensity intermittent exercise sessions rather than offering an immediate performance boost. Although the aerobic contribution to a short-term sprint is minor, it grows as the velocity is increased more frequently [4]. According to the results of studies, the average VO_2max_ value for female basketball players is about 44 to 54 mL/kg/min, and for male basketball players it is about 50 to 60 mL/kg/min. Although values vary by position, guards tend to have a higher aerobic capacity than centers [5,6].

Weekly micro-cycles of training, taper, competition, and recovery make up a competitive season. Due to involvement in local or international events, elite clubs may play different games during a weekly micro-cycle. The need for players to play two to three games each week increases the load placed on them, increasing the chance of injury and performance degradation due to tiredness, muscular damage, and/or inflammation [7]. Acute-phase inflammatory responses characterized by phagocyte infiltration into muscle, free radical generation, and increased cytokines and other inflammatory chemicals are linked to exercise-induced muscle injury. Tumor necrosis factor-alpha (TNF-α), interleukin 1-ß (IL-1-ß), and CRP are only a few of the blood biomarkers that have been employed as indicators of systemic inflammation [8,9]. The acute phase reaction to tissue injury is similar to the inflammatory response. Several cytokines implicated in inflammation, such as TNF-α, IL-1-ß, IL-6, and IL-10, are related to increased plasma levels after strenuous exercise [10].

The HPA axis is hypothesized to be stimulated by IL-6 as part of a natural negative feedback loop that enhances cortisol (an anti-inflammatory mediator) production from the adrenal cortex, while inhibiting pro-inflammatory cytokine release [9]. Furthermore, IL-6 has recently been linked to glucose homeostasis by indicating low muscle glycogen levels. Reduced glycogen availability, calcium homeostasis alterations, and increased reactive oxygen species production can activate transcription factors controlling IL-6 synthesis. Increased circulating IL-6 stimulates the creation of acute-phase proteins like C-Reactive Protein (CRP) in hepatocytes during inflammation [8]. Cortisol and cytokines cause muscle proteins to break down, releasing amino acids into the bloodstream, prompting hepatocytes to absorb these amino acids and synthesize new acute-phase proteins. Cortisol and IL-6 peaks in the circulation preceded CRP peaks in the current investigation, confirming similar findings. CRP elevation has been linked to monocyte activation and the production of adhesion molecules that attract leukocytes. Histidine is a necessary amino acid for protein synthesis in the liver, muscle glycogen storage, and tissue repair, all of which help to lower inflammatory markers like IL-6 and CRP [11]. Carnosine has also been shown to have antioxidant qualities, as well as stimulating effects on the immune system and neurotransmitters (L-carnosine lowers the neural activities of sympathetic nerves and enhances those of parasympathetic nerves) [12].

In basketball, there are anthropometric and bio-motor components. The anthropometric component measures the human body’s composition and describes its proportions. Body weight, height, and fat thickness are all factors in the anthropometric component of basketball [13]. The five basic bio-motor abilities are strength, endurance, speed, flexibility, and coordination. Each exercise in training will tend to develop a particular motor ability. For example, when an exercise load is maximal, it is a strength exercise [14]. Basketball is a team activity that requires much physical fitness and strengthens many bio-motor abilities. Examples of bio-motor components in basketball include speed, agility, strength, coordination, and endurance [13]. Supplementing with β-alanine has been viewed as a potential ergogenic enhancer, mainly for high-intensity sports like basketball, because it is a precursor to carnosine, the most effective intramuscular buffer [15]. Β-alanine supplementation is an unnecessary amino acid that combines with the amino acid histidine to produce carnosine [16].

According to several studies, strenuous exercise increases the expression of various pro- and anti-inflammatory cytokines in the skeletal muscle and blood [17,18]. IL6 expression is up-regulated during the exercise-induced inflammatory process and has an anti-inflammatory effect, causing the release of anti-inflammatory interleukins such as IL10 and reducing TNF-α production [19]. Exogenous supplementation of β-alanine has been proven in vitro and in animal models to raise muscle carnosine concentration [20]. Also, they showed that consumption of β-alanine resulted in a considerable rise in carnosine levels in skeletal muscles, which was linked to better exercise performance [20].

This study aimed to evaluate if 8 weeks of β-alanine supplementation (6.4 g/d) had any effect on the synthesis of exercise-induced cytokines, particularly CRP and IL-6, in elite male basketball players compared to a placebo. We also looked into the impact of β-alanine supplementation on body composition and bio-motor ability. Our hypothesis was that a high dose of β-alanine, along with a longer duration of consumption, would improve body composition and bio-motor capacity and affect CRP and IL-6 levels.

## 2. Materials and Methods

### 2.1. Participants

Twenty male basketball players (age: 23 + 0.6 years; body mass: 78.3 + 4.8 kg; height: 185.3 + 5.4 cm, % BF, 15.2 ± 4.8) volunteered for the study and were randomly assigned to receive either beta-alanine (BG, N = 10) or placebo (PG, N = 10). Subjects had not taken any creatine supplement in the three months before the study. The first group consumed a daily BG (6.4 g/d of β-alanine), and the second group consumed a placebo (6.4 g/d of malt dextrin) in a double-blind format. The computer-generated random table did the randomization [21].

All players were preparing for university competitions and had played for over five years. After explaining all procedures, risks, and benefits, each subject gave informed consent to participate in this study. All subjects signed and accepted the informed consent with the recommendations of the Helsinki Declaration for Human Research and the Ethics Committee of the Faculty of Physical Education and Mountain Sports, Transilvania University of Brasov, Romania, protocol code 54 and agreement 55, date of approval 17 January 2022. The participants were involved in regular basketball training three months before the study. The criteria for selecting and including individuals in this research were not to use additional nutritional supplements and not to consume anabolic steroids or other anabolic agents known to increase performance during the last year. Screening for steroid use, addition, and supplementation was accomplished via a health questionnaire during subject recruitment.

### 2.2. Sample Size

We determined the design’s power and sample size using the statistical approach examined in this study with G-Power software (University of Düsseldorf, Dusseldorf, Germany). Among them were the following [22]: The achieved power was calculated using the a priori and F tests; ANOVA: repeated measurements, within-between interaction analysis, number of groups = 2, number of measures = 6, error probability for α = 0.05, error probability for 1-β = 0.80, and effect size at minimum level = 0.25. In the study with 20 participants, there is an 84.2% (actual power) likelihood of effectively rejecting the null hypothesis that there is no difference in the variables.

### 2.3. Experimental Approach to the Problem

The present study was of an independent group with a pre-and post-test of a semi-experimental design. The participants were randomly divided into two groups based on their specific positions, and players from each position in the game were equally present in both groups. Participants consumed 6.4 g/day (i.e., beta-alanine or maltodextrin) with 300 mL water approximately 1 h prior to training and every morning. Players were assessed for their fitness status and blood sampling two times. The first was assessed in the week before preparing for the match and the second was after 8-weeks. For each period, the players were assessed for three consecutive days. The first day included assessments of anthropometric and body composition (e.g., height, body mass, and body fat), and blood tests. On the second day, anaerobic power was assessed with lactate measurement. Finally, on the third day, the aerobic power test was conducted. Testing days were performed for each participant under a similar environmental condition (21–23 °C temperature and 50% humidity) simultaneously within two-day physical fitness tests [1]. At all sessions, the test had time between 3:30 p.m. and 6:30 p.m. All players presented individual wellness questionnaires before the start of each training session and reported a training load 30 min after each training session. Then, each training load was calculated with the training time. Participants recorded their nutrition for two full days and delivered it to the researchers before the pre-test.

### 2.4. Measurements

#### 2.4.1. Blood Sampling

For analyzing the biochemical variables, the blood taking took place after 12–14 h of fasting during two stages (before and after 8 weeks). In the first stage, the examinees were asked to perform exercises two days before the test. They then attended the medical diagnostic laboratory. The temperature and time of the test were recorded to maintain the conditions in the next stage. Blood was collected directly from the forearm vein by a nurse. After collecting 10 mL of blood, all samples were centrifuged at 3000 rpm for 10 min, and 0.1 mL of the separated supernatant was collected. This process has been analyzed in the laboratory below −80 degrees Celsius. All research variables collected were measured and analyzed by the Gyeonggi-do company.

After this stage, the subjects under study were under specialized exercise for four weeks and were invited again to the laboratory to give blood, as in the first stage, after passing the desirable time and 24 h after the last exercise session. 1L-6 and CRP were tested.

Using an ELISA kit, CRP levels were measured by a nephelometry method (manufactured by the Binding site in the UK). The minimal operating sensitivity of the processor and the kit was 0.04 mg/dL, and the coefficient of variation between and within the processing was 5 and 4/7%, respectively.

IL-6 levels were measured by an immunometric method using an ELISA kit (manufactured by the BioVendor, Heidelberg, Germany). The minimal operating sensitivity of the processor and the kit was 0.92 pg/mL, and the coefficient of variation between and within the processing was 3/4 and 5/2%, respectively.

#### 2.4.2. Body Composition (Body Fat%)

Body composition was analyzed using the InBody720 system (Biospace, Seoul, Korea). The precise body composition analyzer “InBody720” uses 30 Impedance Measurements by using six different frequencies (1 kHz, 5 kHz, 50 kHz, 250 kHz, 500 kHz, 1000 kHz) at every 5 segments (Right Arm, Left Arm, Trunk, Right Leg, Left Leg). Using the 8-point tactile electrode method, InBody measures body composition by segment, and it has body composition analyzing technology that does not resort to empirical estimation such as gender or age [23,24]. Tests were always conducted at the same time of day. All measurements were performed by an expert with five years of background in this area. All anthropometric and body composition measurements were taken in the morning.

#### 2.4.3. Dosage and Supplement Administration

Considering the effective strategies of beta-alanine supplementation (4.8–6.4 g/day) in previous studies, the participants were instructed to consume 6.4 g/day of beta-alanine or maltodextrin supplements for eight weeks [25]. A capsule with 100% purity was provided for consumption. The GNC American company obtained the β-alanine supplement CarnoSyn [26,27,28,29]. The supplement and placebo were in capsule form and were similar in appearance. The participants were randomly assigned in a double-blind manner to receive either β-Alanine or placebo. The PG consumed maltodextrin capsules (Samyang Genex, Seoul, Korea) in the same manner as the BG.

No other side effects (skin tingling, gastrointestinal discomfort) or weight gain were reported by those individuals supplemented with βA, and subjects in the PL group reported no side effects.

#### 2.4.4. Countermovement Jump

The CMJ was utilized to evaluate lower-body power [30,31]. At that point, a standardized warm-up of 10 to 15 min of running was taken after five to six sprint-specific drills, one or two CMJs, even bounds, and vertical bounces. This was followed by one or two trial hops for testing familiarization. Members stood within the center of the contact tangle with hands on the hips. They were taught to plummet quickly until a knee angle of roughly 90° was accomplished, and after that, hop vertically with high speed. Five minutes of rest were given between attempts, and the best execution was recorded in centimeters [32]. Within the CMJ the intra-class relationship (ICC) was 0.96.

#### 2.4.5. Anaerobic Test

A running-based anaerobic sprint test (RAST) was used to measure anaerobic power. For this test, two photocells were based 35 m from each other. Each participant had to run at a maximum speed of six repetitions and take 10-s rest between each repetition. After performing a warm-up with the exact instructions as other tests, the participants began running at maximum speed. After crossing the 35-m line, they rested for 10 s (seconds were counted aloud by the investigator), immediately started again at the end of ten seconds, and this process continued for six repetitions. The records of each participant were calculated with the following formulas and considered for anaerobic power variables: RAST of peak power (RPP) = the highest value; RAST of minimum power (RMP) = the lowest value; RAST of average power (RAP) = sum of all six values ÷ 6; and RAST of Fatigue Index (RFI) = (Maximum power − Minimum power) ÷ Total time for the six sprints.

#### 2.4.6. Lactate Measurements

Blood lactate was measured using a lactometer device immediately after 10 min of training. Lactate was determined using a Lactate Pro 2 device, a palm-sized blood lactate test meter that quickly measures lactate from a small blood sampling (only 0.3 µL). Speedy measurements are completed in just 15 s, offering high performance in a small size.

#### 2.4.7. VO_2max_ Measurements

Bruce’s test measured the maximum oxygen consumption (VO_2max_). The treadmill is started at 2.74 km/h (1.7 mph) and a gradient (or incline) of 10%. At three-minute intervals, the incline of the treadmill increases by 2%, and the speed increases as the guideline. The test should be stopped when the subject cannot continue due to fatigue or pain, or due to many other medical indications. The subjects’ heart rate was also recorded as the heart rate when they stopped test. The device is made in Italy.

#### 2.4.8. Control of Food Intake

Forty-eight hours before and after the test, the Food Frequency Questionnaire (FFQ) was given to the subjects, and they were asked to record their food intake during this period. Dietary monitoring procedures, energy, and macronutrient intake associated with this sample have been previously described in detail. In a nutshell, individuals met with a nutritionist who gave them dietary recommendations that delivered 1.55 times their basal metabolic rate in calories. Players used to eat the same items for 72 h before each measure stages and keep track of their intake. To measure compliance, total calorie and macronutrient intake was measured (Table 1) with Nutrition 4 version 3.5.2 software [10].

### 2.5. Monitoring Internal Training Loads

Internal training loads in players were used to assess and control exercise pressure, with a 0–10-scale rating of perceived exertion (RPE, Borg’s CR-10scale). Each player’s RPE was collected at the end of each specific training to ensure that the perceived effort referred to the training only. In this study, a printed CR-10 scale modified by Foster et al. was used to assist the players in making their responses. All participants of this study were familiarized with this modified scale for RPE before the commencement of this study [33,34].

#### Wellness Measures

Subjective wellness measures were collected using a modified psychological questionnaire based on the recommendations of Hooper and Mackinnon (1995) used by previous studies. The questionnaire comprises subsets of perceived sleep quality, stress, muscle fatigue, and soreness, with each question scored on a 7-point scale (with ‘1′ and ‘7′ representing ‘very good’ and ‘poor’ wellness ratings, respectively) [35]. Overall wellness (Hooper index) was determined by summing the four scores.

### 2.6. Statistical Methods

The Shapiro-Wilk and Leven’s tests were used to evaluate the normality and homogeneity of variances to analyze data. All data are reported as means ± standard deviations. Changes between the pre and post-test were assessed using a repeated-measures analysis of variance (ANOVA), followed by the Bonferroni post-hoc test for pairwise comparisons. Partial eta-square (ηp^2^) was calculated as the effect size of the repeated-measures ANOVA. If the variable was not statistically normal, the Kruskal-Wallis H test was used to analyze the intergroup differences. To calculate the percentage of changes, the post-test average value is subtracted from the pre-test value and is listed as a percentage in the table.
**Percentage Change** (**%**) **=** [(post-test value − pre-test value)/pre-test value] × 100

Furthermore, the effect size of Hedge’s g (95% CI) was reported. The Hopkins threshold was used to calculate the effect size as follows: <0.2 = trivial, 0.2 to 0.6 = small, >0.6 to 1.2 = medium, >1.2 to 2.0 = large, >2.0 to 4.0 = very large, and >4.0, almost perfect. The significance level was considered at *p* ≤ 0.05 [36].

## 3. Results

There were significant (*p*  =  0.003, f  =  11.676, ηp^2^  =  0.393) main effects on time for body mass, but there was no significant group by time interaction (*p* =  0.57, f  =  0.324, ηp^2^  =  0.018). It was the same for fat, as there were significant (*p*  =  0.003, f  =  16.764, ηp^2^  =  0.482) main effects of time, but there was no significant group by time interaction (*p* =  0.11, f  =  2.767, ηp^2^  =  0.133) (Figure 1).

According to the results presented in Table 2, there were significant main effects of time (*p*  =  0.01, f  =  14.089, ηp^2^  =  0.439) and group by time interaction (*p* = 0.01, f  =  15.346, ηp^2^  =  0.460) for IL-6. Post hoc analysis showed that IL-6 was significantly less at the post-test compared to the pre-test in BG (*p*  <  0.001), but for PG there was no significant difference (*p*  >  0.05) between periods. There were significant main effects of time (*p*  =  0.001, f  =  29.569, ηp^2^  =  0.622) and group-by-time interaction (*p* = 0.001, f  =  260.431, ηp^2^  =  0.935) for CRP. Post hoc analysis demonstrated that CRP was significantly less in the post-test compared to pre-test in BG (*p*  <  0.001), while PG was more significant in the post-test compared to the pre-test (*p*  <  0.001). Percent changes in CRP between pre-and post-test were significantly (*p* = 0.001) greater in PG than BG (0.6% > −0.12%).

There were no significant (*p* =  0.66, f  =  0.19, ηp^2^  =  0.01) main effects of time for lactate, and there was no significant group by time interaction (*p*  =  0.54, f  =  0.389, ηp^2^  =  0.021).

According to the results presented in Table 3, there were significant (*p*  =  0.001, f  =  18.766, ηp^2^  =  0.510) main effects of time, but there was no significant group by time interaction (*p* =  0.08, f  =  3.447, ηp^2^  =  0.191) for VO_2max_. There were significant (*p*  =  0.046, f  =  4.583, ηp^2^  =  0.203) main effects of time for heart rate (Table 2), but there was no significant group by time interaction (*p* =  0.231, f  =  1.536, ηp^2^  =  0.085).

There were significant (*p*  =  0.001, f  =  30.502, ηp^2^  =  0.629) main effects of time for RPP, and there was a significant group by time interaction (*p* =  0.045, f  =  4.632, ηp^2^  =  0.205). Post hoc analysis revealed that RPP was significantly (*p*  <  0.01) higher in post-test than pre-test for both groups. Percent changes in RPP between pre-and post-test were significantly (*p* = 0.001) greater in BG than PG (18.9% > 8.3%).

There were significant (*p*  =  0.012, f  =  7.773, ηp^2^  =  0.302) main effects of time for RMP, but there was no significant group by time interaction (*p* =  0.09, f  =  3.087, ηp^2^  =  0.172).

There were significant (*p*  =  0.006, f  =  9.702, ηp^2^  =  0.350) main effects of time for RAP, but there was no significant group by time interaction (*p* =  0.109, f  =  2.839, ηp^2^  =  0.158).

There were significant (*p*  =  0.001, f  =  207.172, ηp^2^  =  0.920) main effects of time for fatigue, but there was no significant group by time interaction (*p* =  0.184, f  =  1.911, ηp^2^  =  0.109). There were significant (*p*  =  0.001, f  =  52.602, ηp^2^  =  0.745) main effects of time for CMJ, but there was no significant group by time interaction (*p* =  0.380, f  =  0.811, ηp^2^  =  0.045).

## 4. Discussion

Basketball has gained enormous popularity worldwide due to its fascinating aspects as a team sport that includes high-intensity activity (sprinting, running, and leaping) phases [1]. When these demands are required to meet the physical and physiological needs of the players, as well as to prevent injuries and deal with inflammation and tissue damage, supplementation with some substances is needed [37]. The goal of the recent study was to investigate the effect of β-alanine supplementation on the state of the pro-inflammatory cytokines CRP and IL-6 as well as the body composition and bio-motor ability of male university basketball players. We hypothesized that 8 weeks of β-alanine supplementation reduces pro-inflammatory cytokines. The results confirmed that 8 weeks of β-alanine supplementation prevented an increase in pro-inflammatory cytokines. In addition, it increased the anaerobic power of trained basketball players. Although both groups reported equal internal workloads, similar energy and macronutrient intakes, and comparable degrees of exhaustion, stress, and sleep, as previously documented, these alterations took place [10].

In the present study, CRP and IL-6 decreased significantly in BG; according to studies, β-alanine supplementation is a non-essential amino acid that combines with the amino acid histidine to produce carnosine, and carnosine significantly reduces the increase in the inflammatory cytokine IL6 [38]. However, IL-6 increased in PG, but this increase was not statistically significant. In addition, there was a significant increase in CRP in PG in male basketball players of the university team. These changes in the basic indicators of pro-inflammatory status in athletes point to the relationship between β-alanine supplementation and the reduction of pro-inflammatory cytokines. The increase of CRP and IL-6 in the PG group with the findings of Kinoshita et al. (2013), who examined the status of pro-inflammatory cytokines in wheelchair basketball players showed that playing basketball caused a significant increase in CRP and IL-6 in male basketball players [39]. The findings of a recent study on the reduction of CRP and IL-6 levels in the BG Group by Dicker are in alignment with the findings of Su-Yeon Jin et al. [40]. They investigated the effects of 4 Weeks of Β-alanine intake on inflammatory cytokines after 10 km long distance running exercise in male students and showed that taking 4 weeks of Β-alanine supplements reduced the level of inflammatory cytokines. In study participants, it effectively maintains immune system function that is temporarily reduced after prolonged exercise [40]. In the current study, β-alanine supplementation was employed for the first time in basketball players, and results suggest that it may be a valuable nutritional tactic for reducing inflammation and promoting faster workout recovery [10].

The measured data about body composition (body mass and fat) in both groups of subjects decreased after eight weeks, but these changes were insignificant. It seems that the slight weight loss was due to training during this period and is not the effect of β-alanine supplementation. Kern and Robinson (2010) investigated the effects of β-alanine supplementation on performance and body composition in collegiate wrestlers and football players. They found that after eight weeks of β-alanine supplementation at 4 g per day, wrestlers in both the β-alanine and PG lost body mass. However, the supplement group increased lean mass by 1.1 pounds, while the PG lost lean mass. Also, both football groups gained body mass t in football tests [41]. These findings are not consistent with the findings of recent research. It seems that the reason for the inconsistency of the obtained results is the difference in the type of exercise activity of the subjects and also the difference in the amount of β-alanine supplement consumption. Contrary to the recent study by Kern and Robinson, the amount of β-alanine supplementation in their study was lower than in the current study. Also, in the current study, lean mass was not measured, and it is possible that the weight loss in the β-alanine group was due to a decrease in fat weight, and the subjects in this group showed an increase in lean mass.

In the section related to VO_2max_, lactate concentration, and heart rate, no significant difference was observed between the two groups’ pre-test and post-test. According to studies, β-alanine can increase the synthesis of L-carnosine and reduce the amount of accumulated lactate in the blood through buffering [41]. The results of the recent research did not agree with the results of Jordan et al. (2010). Jordan et al. investigated the effect of β-alanine supplementation on the onset of blood lactate accumulation (OBLA) during treadmill running. This study showed that VO_2max_ increased in the β-alanine group, and cumulative lactate and heart rate decreased in this group [42]. It seems that the difference in the obtained results is due to the difference in the physical fitness of the subjects of the two studies. In a recent study, semi-professional athletes from the university basketball team, who were probably at their best in VO_2max_, lactate concentration, and heart rate due to their high physical fitness, were used, and did not have a bias for further improvement through supplementation.

In the measured parameters related to bio-motor ability, only the data related to RPP were significant in both the BG and PG groups. Furthermore, there was a significant difference between the two groups in the post-test, and BG was significantly higher than PG. The results were consistent with the results of Ribeiro et al. (2020), who examined the effect of β-alanine supplementation on the performance of elite female soccer players. Ribeiro et al. showed that β-alanine supplementation increases the aerobic capacity measured in the RAST test [25].

There are limitations to this study and suggestions for other research directions. First, this study only included a small number of participants, which may be why some critical findings were not obtained. Also, maltodextrin (like all sugar compounds) is known to have a pro-inflammatory effect, so the placebo may have acted as a negative stressor in the control group and this may have influenced the results. Despite telling individuals to eat similar kinds and amounts of food, we cannot determine whether within-subject variations in micronutrient intake altered inflammatory and hematological measures. Second, while we could monitor internal workload and recovery indicators, we could not use tools to assess external workloads, like GPS devices. Future studies must examine the temporal changes in inflammatory markers over time with β-alanine supplementation in greater detail and ascertain whether the alterations in inflammatory cytokines are sustained with extended supplementation periods. Finally, additional research is necessary to measure changes in physical performance metrics to better correlate the improvements in inflammation markers and immune cells observed in response to β-alanine supplementation.

## 5. Conclusions

The purpose of this study was to evaluate the effect of 8-week β-alanine supplementation on CRP, IL-6, body composition, and bio-motor abilities in elite male basketball players. We have succeeded based on the result showing a significant decrease in CRP and IL-6 and an increase in anaerobic peak power between the pre-test and post-test, as well as between the BG and PG groups. In addition, the result suggested that beta-alanine supplementation may have improved maximal anaerobic power. Although the other measured factors indicated a relative improvement compared to the pre-test and also compared PG, these changes were not statistically significant, and it was not the same direction based on the aim of this study. These changes in pro-inflammatory cytokines suggest that beta-alanine supplementation may be a suitable nutritional strategy to regulate the immune system, and future research is needed to confirm these results and test the effects of beta-alanine supplementation on other pro-inflammatory cytokines and immune system markers are required.

## Figures and Tables

**Figure 1 ijerph-19-13700-f001:**
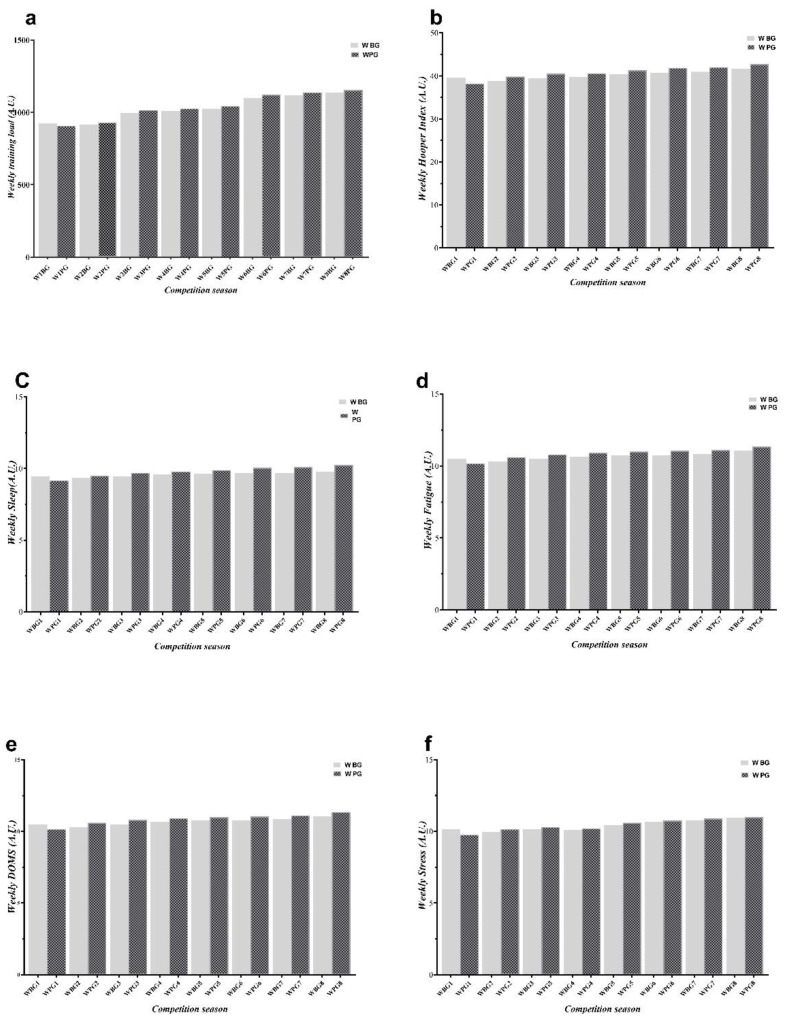
Change in (**a**) Training Load, (**b**) Hooper Index, (**c**) Sleep, (**d**) Fatigue, (**e**) DOMS and (**f**) Stress for 8-week in BG and PG. A.U.: Arbitrary Unit; PG: Placebo group, BG: β-alanine; DOMS: delayed onset muscle soreness.

**Table 1 ijerph-19-13700-t001:** The analysis of the average daily caloric and macronutrient intake.

Variables	Groups	Pre-Season Mean ± SD	Post-Season Mean ± SD	*p*
Carbohydrate (g)	BG	349.30 ± 18.70	355.15 ± 19.8	0.82
	PG	347.18 ± 22.36	342.8 ± 23.5	
Protein (g)	BG	128.26 ± 53.67	125.12 ± 26.5	0.658
	PG	127.65 ± 54.19	119.96 ± 36.8	
Fat (g)	BG	68.32 ± 14.24	69.52 ± 9.23	0.642
	PG	67.86 ± 13.80	65.2 ± 4.9	

Mean (M) ± standard deviation (SD). BG: β-alanine group; PG: placebo group.

**Table 2 ijerph-19-13700-t002:** Changes of cytokines levels and body composition during pre-, mid-, and post-season.

Variables	Groups	Pre-Season M ± SD	Post-Season M ± SD	*p*-Value	Mean Pre-Post Season		Mean Pre-Post Season% Changes	Effect Size Cohens D
					Lower	Upper		
Body mass (kg)	BG	72.45 (0.59)	71.95 (1.3)	0.77	−1.2	0.2	−0.5	−0.8T
	PG	72.25 (0.92)	71.55 (0.92)	0.08	−1.7	0.3	−0.7	−0.5T
Fat (%)	BG	19.35 (0.70)	18.45 (0.49)	0.42	−1.4	−0.3	−0.9	−1.2T
	PG	19.25 (0.42)	18.87 (0.42)	0.72	−0.7	0.01	−0.3	−0.8T
IL-6 (Pg/mL)	BG	5.98 (0.68)	5.42 (0.33)	0.007 *^,#^	−1.07	−0.05	−0.5	−0.8T
	PG	6.14 (0.42)	6.15 (0.31)	0.21	−0.34	0.36	0.01	0.02T
CRP (mg/L)	BG	0.83 (0.08)	0.70 (0.03)	≤0.01 *^,#^	−0.14	−0.1	−0.12	−15.06T
	PG	0.84 (0.09)	0.90 (0.01)	≤0.01 *^,#^	0.05	0.07	0.6	6.4P
Lactate (mg/L)	BG	4.28 (0.28)	4.25 (0.21)	0.16	−0.2	0.2	−0.03	−0.1T
	PG	4.47 (0.21)	4.55 (0.22)	0.40	−0.1	0.2	0.08	0.38S
Heart Rate (BMP)	BG	194.8 (5.39)	195.2 (5.05)	0.34	−4.5	5.3	0.4	0.07T
	PG	193.4 (4.83)	194.9 (4.35)	0.91	−2.8	5.8	1.5	0.3S

Mean (M) ± standard deviation (SD). PG: Placebo group; BG: β-alanine group; * Represents a statistically significant difference compared Pre-test to Post-test; ^#^ Represents a statistically significant difference compared to the PG (exact *p*-value is reported in the text). Cohen’s D was interpreted as T trivial, S small, M medium, L large, V very large, and P almost perfect.

**Table 3 ijerph-19-13700-t003:** Changes of bio-motor abilities during pre- mid- and post-season.

Variables	Groups	Pre-Season M ± SD	Post-Season M ± SD	*p*-Value	Mean Pre-Post Season		Mean Pre-Post Season % Changes	Effect Size Cohens D
					Lower	Upper		
VO_2max_ (mL·kg^−1^·min^−1^)	BG	58.1 (0.87)	60.1 (1.37)	0.007	−3.0	−0.920	−2.000	−1.4T
	PG	58.5 (0.52)	59.3 (0.82)	0.22	−1.4	−0.151	−0.800	−0.94T
Fatigue	BG	43.2 (1.23)	38.4 (1.34)	0.03	3.6	6.0	4.8	3.5V
	PG	43.3 (1.25)	39.3 (0.94)	0.03	2.9	5.0	4.0	4.2P
RPP (w)	BG	862.1 (33.3)	881 (45.7)	0.002 *^,#^	−56.5	18.7	−18.9	−0.4T
	PG	850 (5.12)	858.3 (4.73)	0.002 *^,#^	−12.9	−3.66	−8.3	−1.7T
RMP (w)	BG	411.4 (15.1)	425.5 (7.53)	0.243	−25.3	−2.8	−14.1	−2.07T
	PG	410.8 (5.24)	415 (5.35)	0.89	−9.1	0.7	−4.2	−1.6T
RAP (w)	BG	606.3 (4.62)	616 (6.14)	0.13	−14.8	−4.5	−9.7	−0.3T
	PG	606.4 (9.16)	608 (5.66)	0.24	−9.3	4.9	−2.2	−1.5T
CMJ (cm)	BG	42.4 (1.42)	44.9 (0.72)	0.90	−3.6	−1.5	−2.5	−3.5T
	PG	42.2 (0.82)	44.2 (0.68)	0.61	−2.7	−1.3	−2.0	−2.9T

Mean (M) ± standard deviation (SD). PG: Placebo group; BG: β-alanine group; * Represents a statistically significant difference compared Pre-test to Post-test; ^#^ Represents a statistically significant difference compared to the PG (exact *p*-value is reported in the text). Cohen’s D was interpreted as T trivial, V very large, and P almost perfect.

## Data Availability

The datasets generated during and analyzed during the current study are Available online the corresponding author on reasonable request.

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
