# Peer review of "Effect of 8-Week β-Alanine Supplementation on CRP, IL-6, Body Composition, and Bio-Motor Abilities in Elite Male Basketball Players"

_ijerph, 2022, doi:10.3390/ijerph192013700_

Round 1

Reviewer 1 Report

This manuscript evaluated the effects of B-alanine supplementation over 8-weeks in a cohort of high performance male basketball athletes compared to placebo on various indices of performance and inflammatory status. While the manuscript has some merit in its originality, there is extensive English language editing that needs to be done.

Overall comments:

1. The introduction is quite long and could be shortened in a number of places to ensure a clear description of what has been done and what is lacking in the research is done.

2. How was randomization achieved?

3. Blood sampling techniques need to be expanded on more so to be more clear in the description of what occurred with the blood sampling.

4. When were food frequency questionnaires administered? This should be done both pre- and post- in the study design.

5. What were the standardized procedures for body composition (i.e., bioelectrical impedance analysis) measurement that were used?

6. How was the supplement quality tested?

7. The method in which percent change was calculated was confusing and left this reviewer wondering how it was calculated (in tables 2 and 3)?

8. The conclusion states that "...B-alanine supplementation may be a useful nutritional strategy for immune regulation...". This is not supported by this study as immune regulation was not measured only two inflammatory biomarkers.

9. Overall, there is extensive English language editing that is needed in this manuscript. Spelling, grammar, and sentence structure all need significant revision throughout the paper.

Author Response

We have revised all your comment and concern. We would like to take the opportunity and are grateful for your valuable comments. We hope to get an acceptable response from you.

Best regards

Authors

Academic Editor Notes

English language and grammar should be reviewed by a native speaker.

Authors: thank you. A native speaker person reviewed the article.

The presentation of Table 1 should be improved.

In the conclusion section, state the most important result of your work. Do not simply summarize the points already made in the body, but interpret your results at a higher level of abstraction. Show whether or to what extent you have succeeded in addressing the need stated in the Introduction (or objectives).

Authors: Thank you, we have made general changes in this section

R1

This manuscript evaluated the effects of B-alanine supplementation over 8-weeks in a cohort of high performance male basketball athletes compared to placebo on various indices of performance and inflammatory status. While the manuscript has some merit in its originality, there is extensive English language editing that needs to be done.

Authors: The article was corrected by a native language scholar, which can be seen in the article for changes.

Overall comments:

  1. The introduction is quite long and could be shortened in a number of places to ensure a clear description of what has been done and what is lacking in the research is done.

Authors: thank you, we have edited.

  1. How was randomization achieved?

Authors: Thank you, we have added this information in the method section

Twenty male basketball players (age: 23 + 0.6 years; body mass: 78.3 + 4.8 kg; height: 185.3 + 5.4 cm, % BF, 15.2 ± 4.8) volunteered for the study and were randomly assigned to receive either beta-alanine (BG, N = 10) or placebo (PG, N = 10). Subjects had not taken any creatine supplement in the 3 months prior to the study. The computer-generated random table did randomization.

  1. Blood sampling techniques need to be expanded on more so to be more clear in the description of what occurred with the blood sampling.

Authors: Thank you, we explained exactly what we did it.

  1. When were food frequency questionnaires administered? This should be done both pre- and post- in the study design.

Authors: Thank you. We added the necessary description added

  1. What were the standardized procedures for body composition (i.e., bioelectrical impedance analysis) measurement that were used?

Authors: Thank you. We added the necessary description added.

  1. How was the supplement quality tested?

Authors: This supplement was ordered directly from the official representative of the related company and prescribed for the subjects. This company is one of the most crucial supplement companies in the world. We must also inform the respected reviewer that such a case does not seem visible in sports science studies. To confirm this, we present some highly cited and essential articles on this matter below, none of which such an evaluation has been done. Because all the researchers buy directly from the companies or their official agencies and it can be cited and trusted.

Ribeiro R, Duarte B, Guedes da Silva A, Ramos GP, Rossi Picanço A, Penna EM, et al. Short-Duration Beta-Alanine Supplementation Did Not Prevent the Detrimental Effects of an Intense Preparatory Period on Exercise Capacity in Top-Level Female Footballers. Front Nutr. 2020;7.

Milioni F, Redkva PE, Barbieri FA, Zagatto AM. Six weeks of β-alanine supplementation did not enhance repeated-sprint ability or technical performances in young elite basketball players. Nutr Health. 2017;23(2):111–8.

  1. The method in which percent change was calculated was confusing and left this reviewer wondering how it was calculated (in tables 2 and 3)?

Authors: To calculate the percentage of changes, the post-test average value is subtracted from the pre-test value and is listed as a percentage in the table. In the following articles, the authors have also used this method to calculate the percentage of changes.

Percentage Change (%)= [(posttest value - pretest value)/ pretest value]×100

 (https://www.researchgate.net/publication/359300904_Using_Global_Positioning_System_to_Compare_Training_Monotony_and_Training_Strain_of_Starters_and_Non-Starters_across_of_Full-Season_in_Professional_Soccer_Players)

  1. The conclusion states that "...B-alanine supplementation may be a useful nutritional strategy for immune regulation...". This is not supported by this study as immune regulation was not measured only two inflammatory biomarkers.

Authors: Thank you, we have made general changes in this section

  1. Overall, there is extensive English language editing that is needed in this manuscript. Spelling, grammar, and sentence structure all need significant revision throughout the paper.

Authors: Dear reviewer, as we mentioned above, these changes were reviewed by a native language scientist, and the changes were made. All of them can be seen in the article.

Reviewer 2 Report

Turuc et al conducted a study to test the effect of 8-week β-alanine supplementation on CRP, IL-6, body fat, and bio-motor abilities in elite male basketball players. My main concern is that the sample size is too small to detect the difference. The authors need to address this concern. There are some other suggestions too that are listed below.

Abstract: first sentence is confusing, "decrease pro-inflammatory cytokines and body composition" ? what does decrease body composition means? Please clarify.

Introduction: Please condense the introduction section a little bit. For example, lines 96-99 and 111-132 can be taken off. These are just two examples. There are many other places where the information is provided even though not applicable to the study. 

Ln 121-125 can stay but need clarity. There are typos. Move these sentences after Ln 136

Methods: sample size is too small and must be mentioned in the limitations section. Did the authors conduct a pre-recruitment sample size calculations? If not, they can provide post-hoc sample size needed to test the significance. 

Please provide consort flow diagram for subject recruitment

Ln 187 7ml clogged blood? or blood was taken and clogged?

Ln 202 this is a BIA technique? if so, please clarify in this section and add appropriate reference to show that it's a validated technique to obtain body composition. See PMID: 33472346 and add similar reference that proves the validation of BIA. 

Ln 208 add reference

Not sure why for variables in table 2 and 3 ANOVA value was not reported? For ex, was the change in CRP in PG equal, lower, or greater than change with BG? same for RPP.

Results: Table 1. Please add to the legend that this is 2 day food record before the study day.

Also, in methods, add details on how calories were calculated. Which software was used?

Figure 1 is not clear even on zooming in. Must be improved.

Author Response

We have revised all your comment and concern. We would like to take the opportunity and are grateful for your valuable comments. We hope it will be accepted by you for publishing at this stage.

Best regards

Authors

R2

Turuc et al conducted a study to test the effect of 8-week β-alanine supplementation on CRP, IL-6, body fat, and bio-motor abilities in elite male basketball players. My main concern is that the sample size is too small to detect the difference. The authors need to address this concern. There are some other suggestions too that are listed below.

Authors: Dear reviewer, we calculated the sample size and added as a new section, you can see it below. However, in the limitation section, we added and suggested that future studies to consider more subjects, thank you.

2.2. Sample Size

We determined the design's power and sample size using the statistical approach examined of this study with G-Power software (University of Düsseldorf, Dusseldorf, Germany). Among them were the following: The achieved power is calculated using the a priori and F tests; ANOVA: repeated measurements, within-between interaction analysis, number of groups = 2, number of measures = 2, err prob for α = 0.05, err prob for 1-β = 0.80, and effect size at minimum level = 0.35. In the study with 20 participants, there is an 0.84(actual power) likelihood of effectively rejecting the null hypothesis that there is no difference in the variables.

Abstract: first sentence is confusing, "decrease pro-inflammatory cytokines and body composition" ? what does decrease body composition means? Please clarify.

Authors: thank you, the desired sentence was deleted

Introduction: Please condense the introduction section a little bit. For example, lines 96-99 and 111-132 can be taken off. These are just two examples. There are many other places where the information is provided even though not applicable to the study. 

Authors: thank you. We removed additional sentences in the introduction section.

Ln 121-125 can stay but need clarity. There are typos. Move these sentences after Ln 136

Authors: thank you, we edited it.

Methods: sample size is too small and must be mentioned in the limitations section. Did the authors conduct a pre-recruitment sample size calculations? If not, they can provide post-hoc sample size needed to test the significance. 

Authors: We have already responded in the first comment, thank you very much for your nice guild.

Please provide consort flow diagram for subject recruitment

Authors: Dear reviewer, we thank you for your opinion. In this study, the student’s basketball team participated purposefully from the university team and we did not have any omissions in the study. Therefore, presenting this form may mislead the readers. Because there was no difference in the participants from the time of entering the study to the time of data analysis.

Ln 187 7ml clogged blood? or blood was taken and clogged?

Authors: Blood was collected directly from the forearm vein by a professional nurse, and after collecting 10 mL of whole blood, immediately mixed 5 to 10 times. All samples were stored it at room temperature for 30 minutes, spun at 3,000 RPM for 10 minutes using a centrifuge, collected 1.0 mL of the separated supernatant was transferred to a container for transportation, and transported to the laboratory, Gyeonggi-do, below -80 degrees Celsius and analyzed. All of the collected research variables were measured and analyzed at the request of Gyeonggi-do Company.

Ln 202 this is a BIA technique? if so, please clarify in this section and add appropriate reference to show that it's a validated technique to obtain body composition. See PMID: 33472346 and add similar reference that proves the validation of BIA. 

Authors: The precise body composition analyzer “InBody720”, using 30 Impedance Measurements by Using 6 Different Frequencies (1kHz, 5kHz, 50kHz, 250kHz, 500kHz, 1000kHz) at Each 5 Segments (Right Arm, Left Arm, Trunk, Right Leg, Left Leg). Using 8-point tactile electrode method, In Body measures body composition by segment, and it has body composition analyzing technology that does not resort to empirical estimation such as gender or age.

Ln 208 add reference

Authors: thank you, added.

Not sure why for variables in table 2 and 3 ANOVA value was not reported? For ex, was the change in CRP in PG equal, lower, or greater than change with BG? same for RPP.

Authors: dear reviewer, we have added these values to the text. Thank you.

Results: Table 1. Please add to the legend that this is 2 day food record before the study day.

Also, in methods, add details on how calories were calculated. Which software was used?

Authors: thank you, we edited it.

Figure 1 is not clear even on zooming in. Must be improved.

Authors: thank you; Done. To draw these diagrams, Prism software has been used at the highest quality level 1200bpi, and it is impossible to increase the resolution further. Therefore, we will send the photo separately to the journal.

Reviewer 3 Report

Major concerns:

1. Maltodextrin (as all sugary compounds) is known to have a pro-inflammatory action, so the placebo may have acted as a negative stressor in the control group. This may have greatly affected the results

2. References to pre-clinical studies in diabetic mice may not reflect the physiology of young and fit elite athletes. Authors should find more relevant references, if possible, or explain the use of that background

3. Digression on IL8 is not relevant to the paper

4. IL6 has great variability depending on the sampling method, time from collection and sample management. Authors should better explain this in methods, as it is not clear in the present form

5. An extensive english editing is required, as some parts of the paper are not understandable (i.e. paragraph 2.3.6 "lectometer device immediately after 10 min the training": was it tested immediately or after 10 minutes? was it the end of the training?)

6. Figure 1 is confusing, and should be better explained (firstly by noting which color represents which group)

7. Tables should be edited to highlight better the statistically significant differences (a p-value column could be the solution)

Author Response

We have revised all your comment and concern. We would like to take the opportunity and are grateful for your valuable comments. We hope it will be accepted by you for printing at this stage.

Best regards

Authors

R3

Major concerns:

  1. Maltodextrin (as all sugary compounds) is known to have a pro-inflammatory action, so the placebo may have acted as a negative stressor in the control group. This may have greatly affected the results

Authors: dear reviewer, we agree with you. In some studies, this may have been mentioned, but this is not certain, and it is still used as a placebo in many researches, such as the following articles. However, we have added this in the limitations section.
Effects of beta-alanine supplementation on body composition: a GRADE-assessed systematic review and meta-analysis. Damoon Ashtary-Larkya, Reza Bagherib, Matin Ghanavatic, Omid Asbaghid,
Alexei Wonge, Jeffrey R. Stout f and Katsuhiko Suzuki

The Effect of 28 Days of Beta-alanine Supplementation on Exercise Capacity and Insulin Sensitivity in Individuals with Type 2 Diabetes Mellitus: A Randomised, Double-blind and Placebo-controlled Pilot Trial. Nealon RS*, Sukala WR, Coutts RA and Zhou S

Effect of Beta-Alanine With and Without Sodium Bicarbonate on 2,000-m Rowing Performance

  1. References to pre-clinical studies in diabetic mice may not reflect the physiology of young and fit elite athletes. Authors should find more relevant references, if possible, or explain the use of that background

Authors: thank you, the desired sentence was deleted.

  1. Digression on IL8 is not relevant to the paper

Authors: thank you, the desired sentence was deleted

  1. IL6 has great variability depending on the sampling method, time from collection and sample management. Authors should better explain this in methods, as it is not clear in the present form

For analyzing the biochemical variables, the blood taking was done after 12-14 hours of fasting and in two stages (before and after 8-week). In the first stage, the examinees were asked to do with exercise in two days before the test. Then, they attended the medical di-agnostic laboratory. The temperature and time of the test were recorded to maintain the conditions in the next stage. Blood was collected directly from the forearm vein by a pro-fessional nurse, and after collecting 710 mL of whole blood in 7 mL of SST, immediately mixed 5 to 10 times. All samples were stored at room temperature for 30 minutes, spun at 3,000 RPM for 10 minutes using a centrifuge, collected 1.0 mL of the separated supernatant, transferred to a container for transportation, and transported to the laboratory, below -80 degrees Celsius and analyzed. All of the collected research variables were measured and analyzed at the request of Gyeonggi-do Company.

  1. An extensive english editing is required, as some parts of the paper are not understandable (i.e. paragraph 2.3.6 "lectometer device immediately after 10 min the training": was it tested immediately or after 10 minutes? was it the end of the training?)

Authors: Dear reviewer, these changes were reviewed by a native language scientist, and the changes were made. All of them can be seen in the article.

  1. Figure 1 is confusing, and should be better explained (firstly by noting which color represents which group)

Authors: thank you, changed.

  1. Tables should be edited to highlight better the statistically significant differences (a p-value column could be the solution)

Authors: thank you, we revised all tables.

Round 2

Reviewer 1 Report

Thank you for the modifications that were made to the original manuscript that was submitted. While the current manuscript is improved, I have the following concerns with the current iteration:

1. In the introduction there are a couple instances where the authors state that "According to studies..." (pg 2, line 52) and "According to several studies..." (pg 3, line 150) but there are no references included with the statements. Please include references with the aforementioned statements.

2. Pg 4, line 198 - please spell out the "err prob" completely as error probability.

3. Pg 5, line 209 - how were the players divided based on their particular positions?

4. Pg. 5, lines 231-237 - this needs to be explained more fully as currently it is confusing.

5. Pg.6, lines 268-269 - this is an incomplete sentence.

6. Pg. 6, line 293 - what did the authors mean by "most delicate"? This seems like strange language to use in assessing CMJ.

7. Pg. 11, line 407 - what did the authors mean by stating "more excellent"? I think the word to use would be "higher".

Author Response

Dear Editor and Reviewers,

We appreciate your detailed comments, we have carefully reviewed and responded to all your concerns.  Also, we once again submitted the article to the language department of the university for further correction, and there were minor corrections in the English language, which are highlighted in yellow. The rest of the amendments are in green.

Best regards

The authors

R1

Thank you for the modifications that were made to the original manuscript that was submitted. While the current manuscript is improved, I have the following concerns with the current iteration:

  1. In the introduction there are a couple instances where the authors state that "According to studies..." (pg 2, line 52) and "According to several studies..." (pg 3, line 150) but there are no references included with the statements. Please include references with the aforementioned statements.

Authors: thank you, we have added new references accordingly.

  1. Pg 4, line 198 - please spell out the "err prob" completely as error probability.

Authors: thank you, we defined it.

  1. Pg 5, line 209 - how were the players divided based on their particular positions?

Authors: thank you, added.

  1. 5, lines 231-237 - this needs to be explained more fully as currently it is confusing.

Authors: thank you, we have revised this paragraph, thank you.

  1. 6, lines 268-269 - this is an incomplete sentence.

Authors: thank you, we completed it.

  1. 6, line 293 - what did the authors mean by "most delicate"? This seems like strange language to use in assessing CMJ.

Authors: thank you, we corrected it.

  1. 11, line 407 - what did the authors mean by stating "more excellent"? I think the word to use would be "higher".

Authors: thank you, we edited it.

R2

The authors have well addressed my concerns. However, for the table and figure, they must be stand alone. Readers should not have to go back to text to understand them. Thank you for uploading separate file for figure, make sure to increase the font size for both axes. They are still hard to read.

No need for me to review again. I congratulate the authors for completing this study.

Authors: thank you; Done.

R3

The paper has been improved, but the main issue with the impact on maltodestrin placebo has not been inserted in the text as stated in the reply and should be added in the limitations.

Authors: thank you, added.

Reviewer 2 Report

The authors have well addressed my concerns. However, for the table and figure, they must be stand alone. Readers should not have to go back to text to understand them. Thank you for uploading separate file for figure, make sure to increase the font size for both axes. They are still hard to read.

No need for me to review again. I congratulate the authors for completing this study.

Author Response

Dear Editor and Reviewers,

We appreciate your detailed comments, we have carefully reviewed and responded to all your concerns.  Also, we once again submitted the article to the language department of the university for further correction, and there were minor corrections in the English language, which are highlighted in yellow. The rest of the amendments are in green.

Best regards

The authors

R2

The authors have well addressed my concerns. However, for the table and figure, they must be stand alone. Readers should not have to go back to text to understand them. Thank you for uploading separate file for figure, make sure to increase the font size for both axes. They are still hard to read.

No need for me to review again. I congratulate the authors for completing this study.

Authors: thank you; Done.

Reviewer 3 Report

The paper has been improved, but the main issue with the impact on maltodestrin placebo has not been inserted in the text as stated in the reply and should be added in the limitations.

Author Response

Dear Editor and Reviewers,

We appreciate your detailed comments, we have carefully reviewed and responded to all your concerns.  Also, we once again submitted the article to the language department of the university for further correction, and there were minor corrections in the English language, which are highlighted in yellow. The rest of the amendments are in green.

Best regards

The authors

R3

The paper has been improved, but the main issue with the impact on maltodestrin placebo has not been inserted in the text as stated in the reply and should be added in the limitations.

Authors: thank you, added.